# Prioritizing and characterizing functionally relevant genes across human tissues

**Gowthami Somepalli** [1], **Sarthak Sahoo** [2], **Arashdeep Singh** [3],
**Sridhar Hannenhalli** [3]*

**1** Department of Computer Science, University of Maryland, College Park, Maryland, United States of America, **2** Undergraduate program, Indian Institute of Science, Bengaluru, India, **3** Cancer Data Science Lab, National Cancer Institute, National Institutes of Health, Bethesda, Maryland, United States of America

☯ These authors contributed equally to this work.
* sridhar.hannenhalli@nih.gov

**Data Availability Statement:** All analysis in this work is based on public resources as listed in the text. The opensource code for model application is made available on GitHub https://github.com/somepago/fugue.

## Abstract

Knowledge of genes that are critical to a tissue's function remains difficult to ascertain and presents a major bottleneck toward a mechanistic understanding of genotype-phenotype links. Here, we present the first machine learning model–FUGUE–combining transcriptional and network features, to predict tissue-relevant genes across 30 human tissues. FUGUE achieves an average cross-validation auROC of 0.86 and auPRC of 0.50 (expected 0.09). In independent datasets, FUGUE accurately distinguishes tissue or cell type-specific genes, significantly outperforming the conventional metric based on tissue-specific expression alone. Comparison of tissue-relevant transcription factors across tissue recapitulate their developmental relationships. Interestingly, the tissue-relevant genes cluster on the genome within topologically associated domains and furthermore, are highly enriched for differentially expressed genes in the corresponding cancer type. We provide the prioritized gene lists in 30 human tissues and an open-source software to prioritize genes in a novel context given multi-sample transcriptomic data.

## Author summary

While the identity of most human genes is known, their function is far from established. Even more serious, we do not often know whether a gene serves any function in a specific tissue or context. However, we do know which genes are expressed in a context, and in the absence of any further information regarding their functionality, one simply assumes that the genes that are highly or specifically expressed in a context are functional. While this approach is reasonable in many instances, it is far from ideal, and many genes known to be important are neither highly nor specifically expressed in a context. Here, we focus on this very challenge, and investigate various contextual properties of a gene that can be used to ascertain its functionality. Using several gene features that account for their contextual interaction partners, we propose a machine learning approach to ascertain gene functionality. We show that the resulting tool–FUGUE–improves upon the conventional approach, and by prioritizing the most likely functional genes in dozens of human tissues,

**Funding:** G.S is supported by the University of Maryland. S.S. is supported in part by KVPY fellowship awarded by Department of Science and Technology (DST), Government of India. A.S and S.H are supported by the Intramural Research Program of the National Cancer Institute, Center for Cancer Research, NIH. The funders had no role in study design, data collection and analysis, decision to publish, or preparation of the manuscript.

**Competing interests:** The authors have declared that no competing interests exist.

we further find interesting properties of such genes in terms of their genomic organization, their links with cancer and tissue evolution.

## Introduction

While the list of genes in human and model organisms are fairly complete, our knowledge of their function is far from it; even in the highly studied bacterium E. coli, 35% of all genes lack any experimental evidence for function [1]. Functional pleiotropy of genes further complicates the matter. Moreover, while a gene's function is expected to be highly context-specific, the functional annotations in standard databases such as GO (geneontology.org) are often devoid of context. Here, however, we focus on a more basic question, namely, *Is a given gene functionally relevant in a particular context*? The majority of genes are expressed at physiological levels in any given tissue, but it is not clear if they are relevant to the tissue's function. Context-specific effects of mutations prominently support the context-specific functionality of genes; for instance, while BRCA1/2 are expressed in many tissues, their mutations are associated with primarily breast and ovarian cancer [2]. In the absence of additional information, the convention is to focus to genes that are most highly or most specifically expression in a tissue or context, however, this proxy is far from perfect. A fundamental challenge then is, given tens of thousands of genes expressed in a tissue, to identify those that are likely to be relevant to the tissue's function, i.e., inactivation of the gene has a phenotypic effect observable at the tissue level. Addressing this question has broad implications in interpreting the results of genetic association studies and understanding the mechanisms underlying various diseases, including cancer.

Previous works toward identifying functional genes have been either at cellular level or at organism level. Single gene knockouts have been performed at cellular level, either in human cell line, or in bacteria and yeast, to identify essential genes, based on cellular viability and/or growth rate [3–5]. On the other extreme, single-gene knockout mouse lines (for non-lethal genes) have been studied to identify genes affecting a small set of physiological and behavioral phenotypes [6]. While the cell-based studies are limited to identifying genes essential for cell survival and growth, the organism-level studies are limited in scope to predetermined phenotypes. On the other hand, genome-wide association studies (GWAS) can reveal genes associated with a particular trait or disease, followed by a variety of integrative approaches relying on gene expression, pathways, and networks, to prioritize disease genes [7]. However, GWAS does not immediately suggest the tissue(s) mediating the observed associations, except in the cases where the studied trait is unambiguously ascribed to a specific tissue. As such, there are no standard approaches to assess whether a gene is functionally relevant in a tissue, motivating the current study.

One challenge in developing a model to prioritize tissue-relevant genes (TRGs) is the paucity of a 'gold set', *i.e*, a set of gene-tissue pairs where the functional relevance of the gene in the specific tissue is experimentally established. Here we integrate established disease-tissue maps with the disease genes to compile a high-confidence set of tissue-relevant genes across 29 human tissues (Bladder did not have any relevant genes) and develop a machine-learning model—FUGUE, based on several biologically relevant features, to prioritize functionally relevant genes in each tissue. FUGUE exhibits an overall cross-validation accuracy of 0.86 area under the Receiver Operating characteristic Curve (auROC) and 0.50 area under the Precision Recall Curve (auPRC) (expected 0.09), and equally importantly, for a given gene, FUGUE can distinguish the tissues where the gene is functionally relevant. We provide a functionally prioritized list of genes in 30 human tissues. As expected, the TRGs reveal tissue-relevant functions.

In particular, tissue-relevant transcription factors (TRTFs) include well-established lineage-specifying regulators, and interestingly, cross-tissue comparison of TRTFs recapitulates, to a large extent, tissue developmental hierarchy and their structural relationships. We further found that TRGs tend to be significantly clustered on the genome, in particular within topologically associated domains. Finally, we found that the TRGs are enriched among the differentially expressed genes in the corresponding tissue-specific cancers.

Overall, our work (i) presents the first computational model to prioritize functionally relevant genes in a tissue, (ii) provides prioritized genes lists in 30 human tissues, (iii) along with software pipeline to gene prioritization in a novel tissue or context, and (iv) shows a link between tissue-relevant genes and tissue development, structure, and cancer.

## Results

### Overview of the FUGUE approach

Fig 1 illustrates the overall pipeline. We obtained processed and normalized gene expression profiles of 56202 genes across 11690 samples in 30 tissues from GTEx V8 [8]. We integrated (1) a previously reported literature-based tissue-disease mapping [9], (2) disease-gene mapping curated from OMIM (omim.org) and HPO [9,10], and (3) Human Protein Atlas [11], to curate

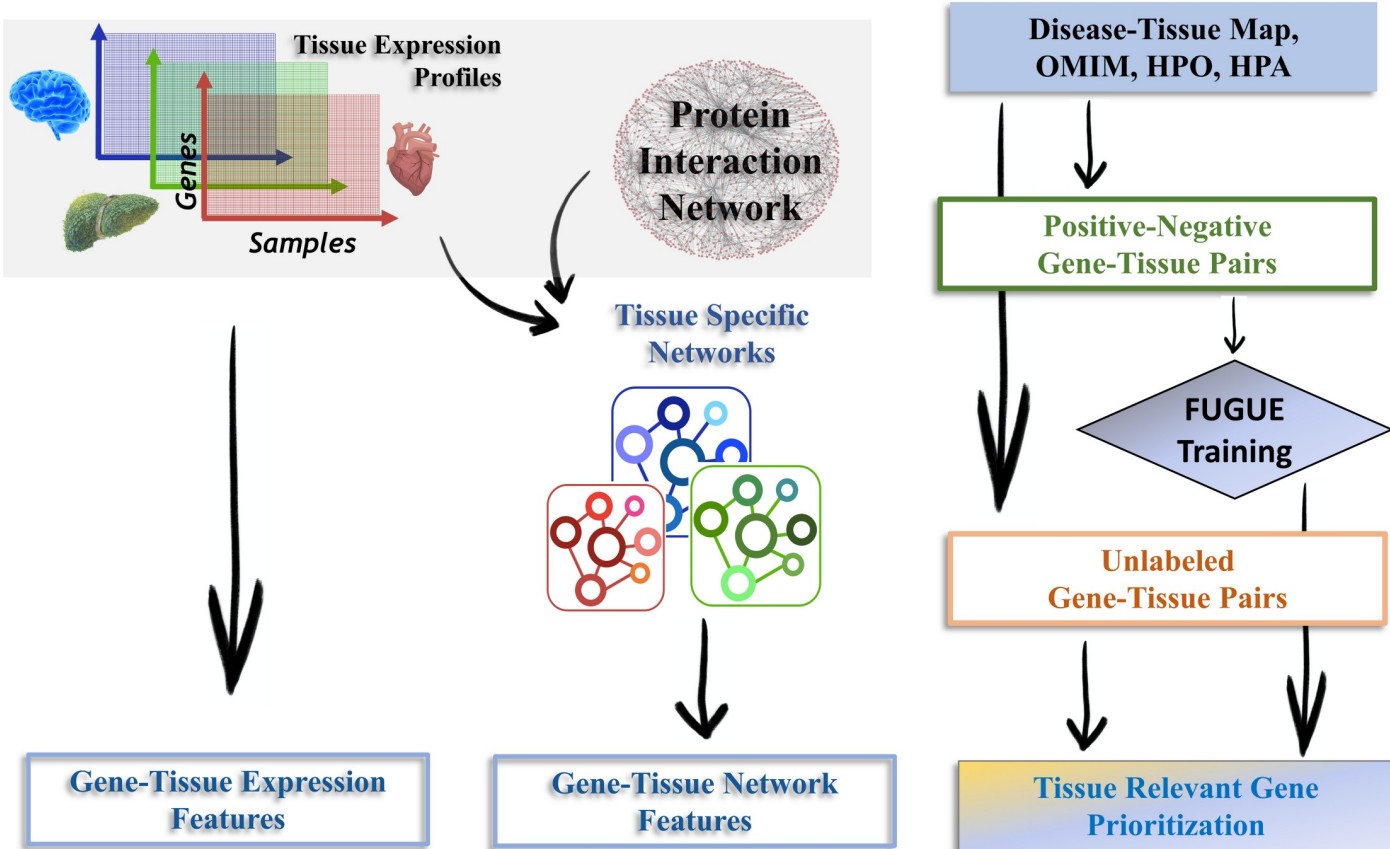

**Fig 1. FUGUE Overview. Left Panel:** Given multi-sample gene expression samples for a tissue, and a human protein interaction network, we derive several expression-based and network-based features for every gene. **Right Panel:** We compile a set of positive and negative gene-tissue pairs by integrating (i) previously curated disease-tissue map, (ii) OMIM database, (iii) HPO database, and (iv) HPA database. Using the compiled gene-tissue pairs we train a XGBOOST model and apply it to prioritize all unlabeled genes in a tissue.

a gold set of 6812 positive gene-tissue pairs and 66250 negative gene-tissue pairs, covering 8325 unique genes across 30 tissues (Methods). We additionally constructed a set of 380393 unlabeled gene-tissue pairs (S1 Data, Methods). S1 Table provides the counts and mean expression of genes in the negative, positive, and the unlabeled genes in each tissue. Next, for 14 biologically motivated features, derived either from tissue-specific gene expression or tissue-specific protein interaction networks (Methods), we assessed the extent to which each feature discriminated between the positive and negative gene-tissue pairs. Finally, combining the features we developed a XGBoost classification model [12] and assessed its performance at three different levels: (1) overall discrimination between positive and the negative gene-tissue pairs, (2) for each gene, discrimination between the relevant tissues and the other tissues, and (3) for each tissue, discrimination between the tissue-relevant genes and the rest.

## A machine learning model to predict a gene's relevance in a tissue

We first assessed each of the 14 features individually and found that for 12 out of 14 features, the feature value was significantly different (Wilcoxon p-value $< = 0.05$) between the positive and the negative gene-tissue pairs (Table 1). The 2 remaining features—NeighborMeanBreadth (mean expression breadth of the protein interaction neighbors of the gene) and NeighborMeanZscore (mean expression Z-score of the protein interaction neighbors of the gene), although not statistically significant across the pooled positive and negative samples, nevertheless exhibited significant difference between positive and negative genes when tested tissue-wise across the tissues (S1 Fig). We additionally quantified each feature's ability to distinguish between pooled sets of the positive and the negative gene-tissue pairs using area under the receiver operating curve (auROC), as well as area under precision-recall curve (auPRC) (Table 1). The auROC ranged from 0.61 to 0.73, and the auPRC ranged from 0.13 to 0.24 across the features; expected auPRC given the size of positive and negative instances is 0.09. Overall, all investigated features can significantly discriminate, albeit to varying degrees, the positive gene-tissue pairs from the negative control.

Given the modest, but significant, discriminative power of individual features, we developed a machine learning tool—FUGUE, by combining all features in a XGBoost (gradient boosted decision tree) model [12] (Methods). We first quantified FUGUE's ability to

**Table 1. Biological features and their potential to discriminate positive and background gene-tissue pairs.** P-value: Wilcoxon test significance comparing the feature values for the positive and the negative gene-tissue pairs. auROC: Area under the receiver operating characteristic curve. auPRC: Area under the precision recall curve.

| Feature | p-value | auPRC | auROC | Neg Mean (N) | Pos Mean (P) | Effect size = P/N |
|---|---|---|---|---|---|---|
| MeanExp | 4.47E-250 | 0.19 | 0.72 | 39.74 | 215.37 | 5.42 |
| SDofExp | 7.29E-177 | 0.16 | 0.68 | 31.49 | 148.56 | 4.72 |
| ZScore | 4.26E-18 | 0.24 | 0.73 | -0.13 | 0.30 | 2.30 |
| MedianExp | 1.54E-265 | 0.18 | 0.72 | 31.57 | 180.90 | 5.73 |
| Breadth | 1.47E-06 | 0.17 | 0.62 | 0.48 | 0.52 | 1.08 |
| CV | 3.54E-154 | 0.13 | 0.63 | 1.36 | 0.68 | 0.50 |
| MAD | 1.38E-250 | 0.17 | 0.70 | 19.87 | 120.41 | 6.06 |
| Centrality_coeff | 1.13E-15 | 0.15 | 0.65 | 0.00 | 0.00 | 1.02 |
| Clustering_coeff | 5.71E-11 | 0.15 | 0.66 | 0.10 | 0.11 | 1.04 |
| Degree (NumOfNeighbors) | 2.26E-43 | 0.18 | 0.69 | 41.51 | 56.01 | 1.35 |
| NumOfKinaseNbs | 6.79E-13 | 0.17 | 0.61 | 1.75 | 2.17 | 1.24 |
| NumOfTFNeighbors | 9.34E-38 | 0.17 | 0.65 | 3.93 | 5.60 | 1.42 |
| NeighborMeanBreadth | 1.00E+00 | 0.16 | 0.65 | 0.51 | 0.43 | 0.83 |
| NeighborMeanZscore | 1.00E+00 | 0.14 | 0.61 | -0.02 | -0.06 | 3.47 |

discriminate between pooled positive and negative gene-tissue pairs. We performed 5-fold cross validation for 100 ensembles, yielding an overall average auROC score of 0.86 and auPRC of 0.50 (5-fold greater than random expectation). In a given tissue, for a gene to be marked as negative, we require that the gene's protein product is not detected in the corresponding tissue in HPA database. This raises a possibility of ascertainment bias in Z-score distribution between the positive and the negative gene-tissue pairs. First, we note (S1 Table) that the genes in N exhibit a range of mean mRNA expression values in the GTEx dataset, and notably, in some tissues the mean expression of genes in N are comparable or even greater than the mean expression in the positive (P) set. To rule out Z-score as a major contributor to our prediction accuracy, we randomly subsampled the negative gene-tissue pairs to match their Z-score distribution to the positive gene-tissue pairs. Even in this matched data, FUGUE achieves a cross-validation auROC of 0.8 (random expectation 0.5) and auPRC of 0.55 (random expectation 0.2). We further assessed the discrimination between the P and unlabeled (U) genes (representing the genes that have detectable proteins in the tissue in HPA). Although a bit lower than the P~N performance, the P~U discrimination accuracy was nevertheless reasonable high with auROC of 0.78 and auPRC observed/expected = 4.6.

Next, we assessed FUGUE's ability to prioritize the functionally relevant genes in each tissue individually (Methods). In each of the 20 tissues (having both positive and negative genes), we quantified the normalized ranks (based on FUGUE score) of the positive and the negative genes; to avoid overfitting, the model was trained on all the tissues except the one being tested. In all 20 tissues, the positive genes were ranked (based on FUGUE scores) significantly higher than the negative genes (p-value < = 0.05); normalized mean ranks of tissue-specific positive and negative genes are shown in Fig 2A. The outlier tissue in Fig 2A having a lower average rank for positive genes is Pituitary gland; upon closer inspection, this tissue has the lowest number (2) of positive genes and 65 negative genes, and thus is not a representative. We further assess merits of an integrative model relative to the conventional Z-score based ranking, we directly compared FUGUE and Z-score ranks for the positive genes in each tissue (excluded from the training set). Of the 25 tissues having positive genes, we found that in 11 tissues FUGUE ranks were better (P < 0.05) than Z-score ranks. Notably, Z-score ranks were not better than FUGUE ranks in any of the tissues; the numbers are 9 out of 13 tissues if we only consider tissues with at least 100 positive genes.

Next, we assessed FUGUE's ability to prioritize the functionally relevant tissues for each gene. For each gene g, we trained the model on all genes except g, and then estimated the score for g in all tissues. We then compared, for each gene, the normalized ranks of the gene in the positive tissues with that in the negative tissues. Fig 2B suggests that across all genes, the ranks in positive tissues are higher than in the negative tissues (paired Wilcoxon test p-value = 4.19e-63). Fig 2C shows the relative contributions of top contributing features (Methods). Z-score is the conventional measure to prioritize tissue-relevant genes [11], and as expected, it is highest contributor to the model. However, notably, Coefficient of Variation (CV), which captures expression variability across samples within a tissue, contributes almost as much, revealing that tissue relevant genes have significantly lower inter-individual variation than other genes. S2 Fig shows substantial variability in Z-scores among the top 5% FUGUE-ranked genes in each tissue. We further assessed the incremental contribution of each feature by assessing the overall auROC for select individual features and various feature groupings (Fig 2D). The figure shows that even though the expression features perform as well as network features, there is a value added in the combination. Overall, FUGUE provides an effective predictive model of tissue-relevant genes. The FUGUE scores for every gene in all 30 tissues are provided in S1 Data. While we have provided the FUGUE scores for all genes, for some of validations below, where appropriate, we use the top 10% highest scoring genes as the nominal tissue relevant genes.

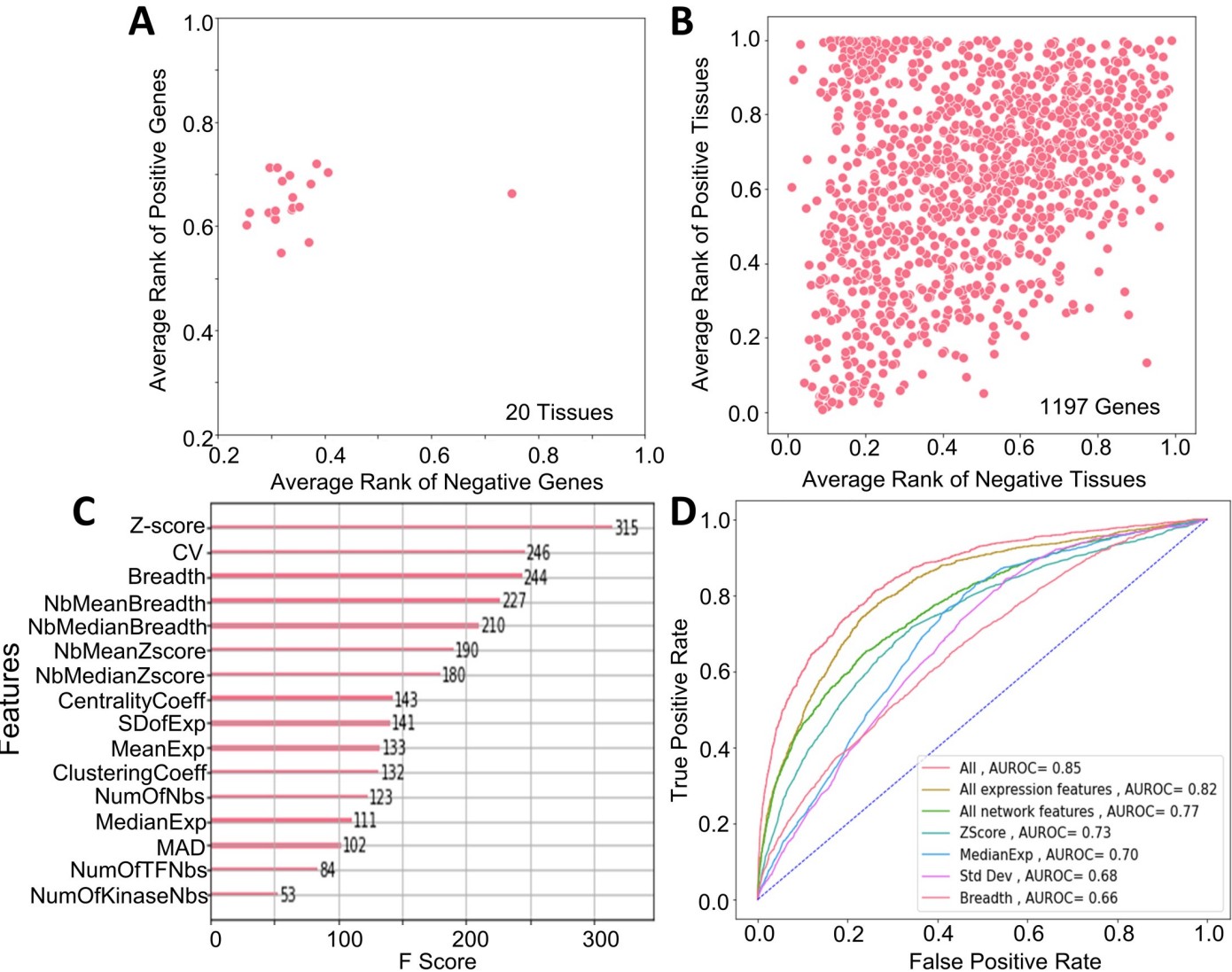

**Fig 2. FUGUE performance. (A)** Positive genes in each tissue is ranked based on a model trained on other tissues. Mean rank of positive genes in each tissue (y-axis) is consistently far higher than the negative genes in that tissue. (x-axis). **(B)** Similar to (A) but here each gene's rank in Positive tissue (y-axis) is compared with the same gene's rank in negative tissues (x-axis); the model training excludes the gene of interest. **(C)** Each feature's importance (F score) estimated by the model is shown. **(D)** ROC for overall cross-validation accuracy for various combinations of features.

## FUGUE validation in independent datasets and utility in mapping complex traits to tissues

Next, we assessed the utility of our tissue-specific FUGUE-prioritized gene lists in independent datasets. First, we compiled from the whole genome mouse knockout study [13], 191 genes whose germline deletion lead to phenotypic aberration that could be unambiguously ascribed to one of the 6 tissues—Heart, Breast, Thyroid, Muscle, Lung, and Ovary (S2 Table). After mapping the mouse genes to their human orthologs [13], we found that for each of the 6 tissues the experimentally identified genes were ranked significantly higher than other genes by FUGUE (Fig 3A; Wilcoxon p-values ~ 0 in all cases). A direct comparison with Z-score revealed that in 2 of the 6 tissues (Breast and Ovary), FUGUE rankings of the tissue-relevant genes were significantly higher (p-value <0.05) than Z-score rankings and the converse was

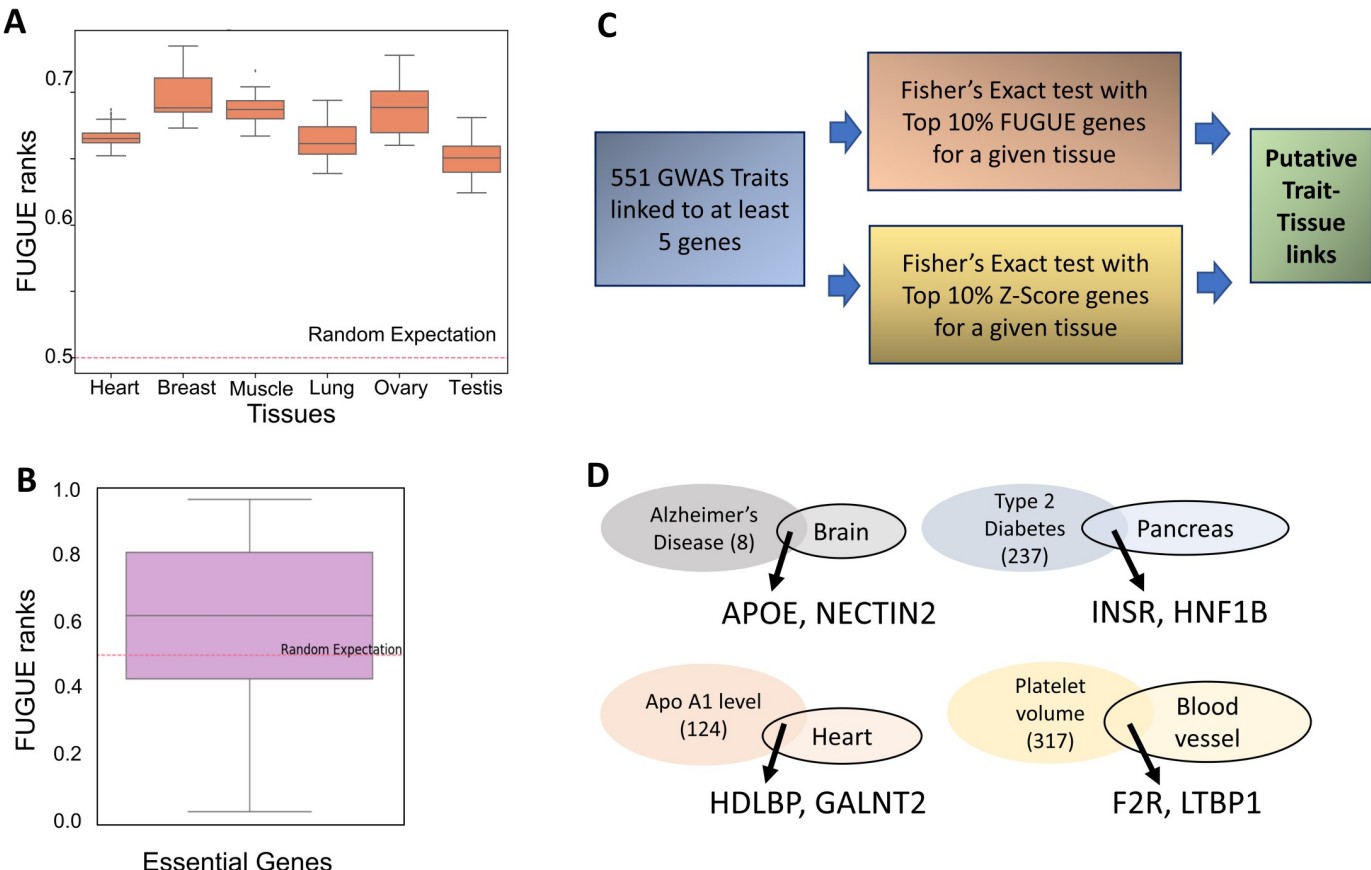

**Fig 3. FUGUE validation and application to trait-tissue mapping. (A)** In 6 tissues where the trait is unambiguously mapped to the tissue based on mouse knockout, the tissue-relevant genes are ranked significantly higher (y-axis) than the random expectation. **(B)** Essential genes in human iPSCs are ranked much higher than random expectation; here the model was trained on GTEx and applied to iPSC, minimizing overfitting. **(C)** Pipeline to map GWAS genes to specific tissues based on the overlap of GWAS genes with the top ranked genes by FUGUE or by Z-score. **(D)** A few examples of trait-tissue mapping uniquely revealed by FUGUE. The trait oval indicates the number of GWAS genes, the tissue oval indicates the top 10% TRGs and a couple of top ranked genes associated to the trait are highlighted. These genes have direct literature evidence for involvement with the trait.

true in 1 tissue (Heart), while in the other 3 tissues (Thyroid, Muscle, and Lung), there was no significant difference between the two. However, in general our result reveals numerous genes that have low relative expression in a tissue and would therefore not be deemed interesting based on Z-score but are nevertheless identified by our integrative model as being potentially important for a tissue. Here we highlight a few examples. YME1L1, whose disruption is known to cause cardiac dysfunction [14], has a z-score of -0.87 in heart tissue (S2 Table) but has FUGUE score of 0.92. APC, a DNA repair gene linked to breast cancer [15] has a z-score of -0.3 in breast while a FUGUE score of 0.7. PPP2R1B linked to lung cancer [16] has a z-score of -0.08 in breast while a FUGUE score of 0.7. PUS1, whose deficiency affects muscle morphology in mice [17], has a z-score of -0.78 in muscle but a FUGUE score of 0.94. KRAS, mutations in which cause ovarian carcinoma [18], has a z-score of -0.12 in ovary and FUGUE score of 0.82. In fact, KRAS is involved in several cancers, notably colorectal cancers [19]. Indeed, KRAS normalized rank was above 0.9 in 13 tissues, including colon where it was 0.97. TGF-beta functions in gonadal development [20]. It's receptor TGFBR2 has a z-score of -0.92 in testis while a FUGUE score of 0.82. These examples, and many more in our prioritized lists, strongly underscore the value of our integrative model.

Next, we applied FUGUE (trained only on GTEx tissues) to 317 human Embryonic Stem Cell (hESC) transcriptome [21](Methods), and tested whether the 500 genes deemed to be most essential, based on CRISPR-Cas9 knockout in human pluripotent stem cells (hPSC) [22], were ranked higher than expected. As shown in Fig 3B, this is indeed the case (Wilcoxon p-value = 3.4E-19). However, Z-score ranks of the essential genes was higher than the FUGUE ranks. We assessed whether superior performance of Z-score might be because of a small fraction of genes that are highly specific to hPSC (i.e., highest Z-scoring genes). We found that if we exclude the genes with top 5% Z-scores, then the FUGUE ranking is higher than Z-score rankings (P = 2E-4) for the remaining genes, and if we exclude top 10% genes with highest Z-score then the signal gets even stronger (1E-10). However, it is also possible that our FUGUE model trained on differentiated tissues is not perfectly suited to ESC. We therefore further tested the relative merits of an integrative model trained on hESC. We assessed the 5-fold cross-validation accuracy of FUGUE on 317 hESC transcriptomic samples and found that an integrative model achieved an auROC of 0.78 ± 0.02 and auPRC of 0.35 ± 0.04, compared to Z-score which achieved an auROC of 0.67 ± 0.02 and auPRC of 0.22 ± 0.03. This confirms the contributions of additional transcriptomic and network features in prioritizing the essential genes in hESC.

The above validations suggest that while FUGUE performs better than Z-score alone in a cross-validation fashion, in a new context, an independently trained FUGUE model is complementary to Z-score based ranking and does not necessarily supersede it. Next, we applied the FUGUE as well as Z-score based ranking to map complex diseases and traits to specific tissues based on the premise that the genes associated with trait based on GWAS studies ought to be enriched among the top ranked genes in the relevant tissue(s). For a comprehensive list of 551 traits from the GWAS catalog [23] having at least 5 associated genes, in each of the 30 tissues, we assessed whether the GWAS-linked genes significantly overlapped (odds ratio > 1.5 and Fisher test p-value <0.05) with the top 10% FUGUE or top 10% Z-score ranked genes in the tissue. On average per tissue Z-score ranking revealed 34 traits and FUGUE revealed 30 traits, 29% of which overlapped with Z-score (S3 Table). While some of the traits revealed individually by FUGUE or Z-score could be directly linked to corresponding tissue function either under homeostatic conditions or in diseases, in many cases the specific contribution of the tissue to the trait is difficult to rationalize and likely false positives. However, the intersection of the mapping by FUGUE and Z-score are often supported by known biology. Next, we discuss a few cases that are uniquely revealed by FUGUE ranking or by both rankings.

In Brain, Alzheimer's disease and cognitive ability were uniquely revealed by FUGUE, while Neuroticism was revealed by both. In Liver, both gene lists were sensitive enough to capture numerous metabolic functions. In Pancreas, FUGUE uniquely identified Type II diabetes. Calcium level was revealed by FUGUE and Z-Score (>13 odds ratio), in line with essential role of calcium levels in regulation and release of insulin by the pancreas and in its implication during the development of type II diabetes [24]. In Heart, physiological traits, such as electrocardiographic traits including PR interval, resting heart rate, QRS duration, P wave duration, as well as atrial fibrillation were associated with both gene lists. Genes linked with Apolipoprotein A1 level were uniquely enriched among top FUGUE genes in heart. Apolipoprotein levels have been implicated in atrial fibrillation and failing human hearts [25,26]. In Lung, asthma, tuberculosis, respiratory diseases, etc were robustly captured by both gene lists. In blood, various traits related to blood cell counts and white blood cell functions were associated with both gene lists. In Ovary, resting TPE interval was enriched for by both gene lists. TPE intervals have been implicated in premature ovarian failure [27]. A seemingly disconnected association uniquely captured by the FUGUE is squamous cell lung carcinoma (SCLC). SCLC frequently metastases to ovary and often the lung and ovarian tumors occur synchronously [28]. In Testis,

counterintuitively, esophageal cancer and glioblastoma are uniquely mapped by FUGUE. We find that as cancer-testis antigens are frequently implicated in both these cancer types [29,30]. In Thyroid, thyroid stimulating hormone levels, creatinine levels and hyperuricemia are identified by both gene lists. These traits are directly related to the function of thyroid tissue and are dysregulated in diseases associated with thyroid gland. In Prostate, prostate cancer was associated with both gene lists. These and several additional examples are listed in S3 Table.

## TRGs recapitulate tissue-relevant functions and developmental and structural tissue relationship

To assess whether TRGs reflect normal tissue functions, we assessed Gene Ontology (GO) functional enrichment among the top 5% TRGs in each tissue, using PANTHER web server [31] with the default background of all human genes, followed by Bonferroni Correction; only GO terms with a family wise error rate < 0.05 were considered. For most tissues, the enriched functions recapitulate the tissue-relevant biological processes (S2 Data). For instance, for the brain tissue, top GO terms identified were overwhelmingly related to functions involving specific biological processes at the synapses (synaptic vessel endocytosis, recycling and localization, regulation of synaptic vessel exocytosis, regulation of neurotransmitter receptor activity, protein and receptor localization to the synapses, etc) along with functions related to ionotropic glutamate receptor signaling pathways and NMDA receptor activity. Furthermore, for the nerve tissue, we specifically found that GO terms for myelination in peripheral nervous system (myelin maintenance and axon ensheathment related functions) and Schwann cell differentiation and development were highly enriched, demonstrating the specificity of the identified TRGs. Similarly, we found for the heart tissue, genes related to cardiac muscle contraction, actin-myosin filament sliding, heart contraction, cardiac muscle cell development, etc. were significantly enriched. Among other tissue specific biological functions, thyroid hormone generation for thyroid gland, pancreas development for the pancreas tissue, actin-myosin filament sliding for muscle tissue, T cell and lymphocyte homeostasis in the blood, chylomicron assembly and remnant clearance along with a vast number of carbohydrate, fatty acid and cholesterol metabolism related GO terms in the liver, etc. were all significantly overrepresented among the top 5% TRGs. All these GO terms, along with numerous others, point to the high level of specificity to tissue relevant functions of the identified TRGs. On the other hand, housekeeping genes perform important functions in many tissues and are therefore expected to be ranked highly in many tissues. Indeed, we found that the average pan-tissue FUGUE scores of known housekeeping genes [32] is significantly higher than the rest of the genes (Wilcoxon p-value ~ 0).

Next, we assessed whether the most relevant regulatory proteins, namely transcription factors, reveal known tissue development and biology. Upon closer inspection of the top TRGs, we found a large number of master regulators highly characteristic of the tissue were scored highly. For instance, PPARG and HNF1A, master regulators for adipogenesis and the hepatic cell fate respectively [33], are among the top TRGs for the adipose and the liver tissues respectively and are found to be implicated in NAFLD [34] and steatosis associated liver cancer [33]. Similarly, insulin is the highest scored gene (FUGUE score: 0.98) in the pancreas tissue. These observations underscore the high degree of specificity of FUGUE identifying functionally relevant tissue specific genes that are also implicated in various diseases.

We further assessed whether the tissue relevant regulators reveal developmental or functional relationships among tissues. Toward this, we quantified similarity between each tissue pair based on their Jaccard index of overlap in their top 20 FUGUE-ranked TRTFs, and performed hierarchical clustering of tissues primarily derived from the endodermal (stomach,

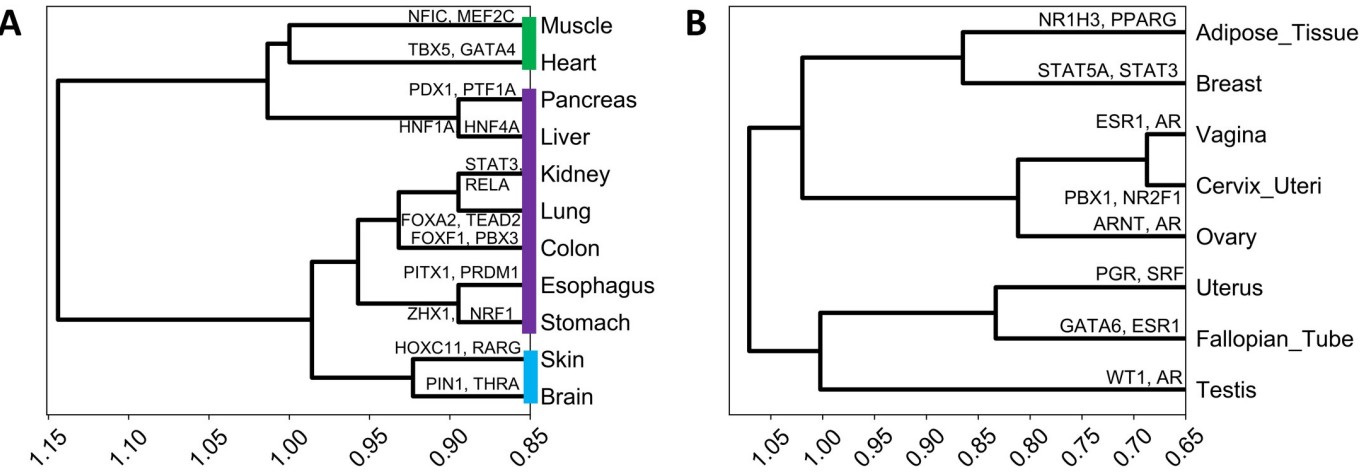

**Fig 4. Top TRTFs capture developmental as well as structural relations among several mature tissues. (A)** Cluster dendrogram of tissues belonging to the endodermal (purple), mesodermal (green) and the ectodermal (blue) lineages. **(B)** Cluster dendrogram of primary and secondary reproductive organs and adipose tissue. As an illustration, we have shown for each tissue, two top TFs previously shown to be involved in the tissue development or homeostasis.

colon, esophagus, lung, kidney, liver and pancreas), mesodermal (muscle and heart) and the ectodermal (brain and skin) lineages [35] (Fig 4A); top 20 TFs for each tissue is provided in S4 Table. Encouragingly, we found that the stomach and esophagus clustered together consistent with the fact they are derived from the foregut [36] and away from colon which is derived from the hindgut [36]. Furthermore, Lung and Kidney clustered together, consistent with their developmental links [37]. However, we found that liver and pancreas, although clustered together consistent with their developmental relationship [38], did not cluster closely with the other endodermal lineage tissues. This is however consistent with potentially parallel development of these two tissues from the gut tube as revealed recently by single cell analysis [39]. Not surprisingly, Heart and Muscle, related tissues of mesodermal lineage [40], clustered together but farther from the endodermal origin tissues. Similarly the brain and skin are clustered together and separately from the mesodermal lineage and the endodermal lineage owing to the fact that both these tissues are largely descendants of the ectodermal lineage [41]. Similarly, we found that overlap among the top TRTFs could also classify the primary and secondary reproductive organs based on their structural proximity (Fig 4B). Intriguingly we found adipose tissue (not a primary/secondary reproductive organ) to cluster together with breast tissue. This may reflect high adipose content in breast tissue. These results strongly suggest that the predicted TRGs provide insights into the tissue-relevant functions and TRTFs capture the developmental as well as structural relations among several mature tissues.

## TRGs organize into genomic clusters and are perturbed in cancer

Functionally and transcriptionally related genes tend to cluster on the genome, both linearly [42] as well as spatially mediated by chromatin structure [43]. We therefore assessed whether, owing to their functional relatedness, top 5% TRGs exhibit linear or spatial clustering. To assess genomic clustering of TRGs in each tissue, we first defined a genomic cluster as a set of five or more genes where the consecutive genes were no more than 500 kb apart. We quantified genomic clustering as the fraction of TRGs covered by the defined clusters and estimated the z-score and an empirical p-value based on chromosome-wise randomized gene sets (Methods). The z-score distribution (Fig 5A) strongly suggests that TRGs tend to cluster on the

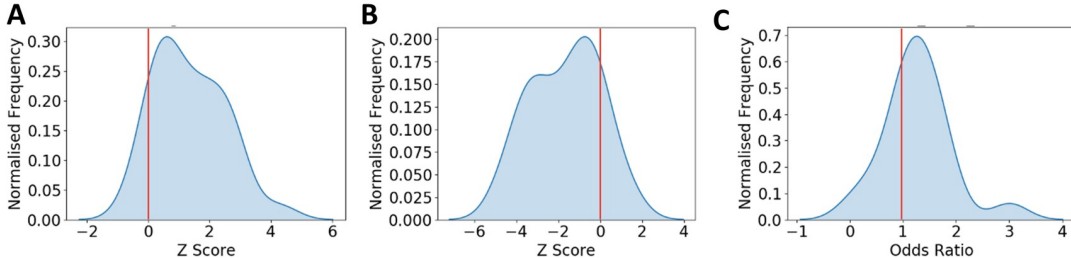

**Fig 5. TRGs cluster both on the linear as well as in the 3D genome and are dysregulated in tissue specific cancer. (A)** Z-score distribution for fraction of TRGs contained in clusters (size > = 5 TRGs) compared to a null distribution along the linear genome (top 5% TRGs considered; 5 or more genes in a cluster no more than 500kb apart) **(B)** Z-score distribution of number of TADs containing more than 5% of TRGs **(C)** Odds ratio distribution for TRGs enrichment in dysregulated genes in cancer types. Red line denotes the expected mean.

genome. Specifically, in 28 out of 30 tissues, clusters cover greater than expected fraction of TRGs (z-score > 0), out of which 11 are significant (p-value < = 0.05; S5 Table). The above trends were qualitatively similar when we define a cluster to be composed of 3 or more genes (S5 Table).

Analysis of spatial chromatin interactions via Hi-C experiments reveal an organization of chromatin into Topologically Associated Domains (TADs) on the linear genome within which disproportionately large fraction of chromatin interactions reside, resulting in a coordinated transcription within TADs [44,45]. We therefore assessed the extent to which TRGs are organized within TADs (Methods). Toward this, we measure the number of TADs covering the TRGs—smaller the number, the greater the association between TRGs and TADs (Methods). As above, we compare this metric against a randomized control, in a tissue-specific fashion, to estimate a z-score and an empirical p-value. In 26 out of 30 tissues we observe a negative Z-score (Fig 5B), 14 of which are significant (p-value < 0.05) (S6 Table). The TADs used for the above analysis are not tissue-specific and in general TADs have been reported to be largely invariable across tissues. To assess tissue-specificity of our observations, we leveraged the recently reported tissue-specific enhancer-gene maps in EpiMap [46]. We obtained the tissue-specific gene-enhancer maps for the 8 tissues that could be unambiguously mapped between EpiMap and our study—Brain, Kidney, Heart, Liver, Lung, Muscle, Pancreas, and Spleen. We expect TRGs to be under greater tissue-specific transcriptional control, i.e., interacting with a greater number of enhancers. We tested this by comparing the number of enhancers linked with TRGs with random gene sets. In 7 of the 8 tissues (except Brain), the tests strongly support our expectation.

Collectively these results suggest that TRGs have a strong propensity to cluster on the linear genome, especially within TADs.

Given the observed potential role of TRGs in tissue development, and previously observed links between development and cancer [47], we assessed the extent to which TRGs are perturbed in the corresponding cancer. Sixteen tissues in our study were mapped to 19 cancer types in TCGA (www.cancer.gov/tcga). For each cancer we obtained the genes that are differentially expressed DEGs) (both over-expressed and under-expressed) in the tumors compared to corresponding control [48]. We tested for overlap between TRGs and DEGs separately for up- and down-regulated DEGs. We found that in 16 out of the 19 cancers, the top 5% TRGs were significantly enriched (Fisher exact test p-value < 0.05) in either up-regulated or down-regulated DEGs (Fig 5C and S7 Table), suggesting that TRGs are more likely to be transcriptionally perturbed during oncogenesis.

## Discussion

Here we present the first machine learning model to assess whether a gene is relevant to a particular tissue's function, based on multiple biologically grounded tissue-specific features of a gene that rely both on expression as well as protein interaction networks. This represents a significant step forward relative to the standard convention of using average expression of gene in a tissue relative to other tissues, i.e., Z-score, to prioritize tissue-relevant genes. However, we note that in an entirely new context, such as hPSC in our study, Z-score remains a potent characteristic of functional gene and should be considered along with an integrative score. While Z-score is indeed revealed in our model as an important feature, notably, low cross-sample variability of the gene in the specific tissue turns out to be highly important. We also show that features derived from tissue projection of the protein-interaction network alone are as effective as the combined expression-derived features, in line with previous work, NetWAS [49], showing that tissue-specific gene networks can identify disease-associated genes more accurately than the GWAS. Gene-centric epigenomic features may further enhance our model accuracy; however, such features are not widely available across all tissues, and furthermore, are reflected to a large extent in the gene expression data.

Lack of sufficient number of positive and negative genes in each tissue has precluded us from building tissue-specific models. However, we have shown that our model performs accurately in a tissue even when that tissue's data is excluded from the model training, suggesting the model captures universal features of TRGs. This is also evident in our independent validations in mouse knockout, as well as hESC cells, that were not used for model training.

Our model prioritizes key transcription factors involved in development or functioning of various tissues, such as HNF4 in liver [50], GATA4 in heart [51], MEF2C in muscle [52], NR2F1 in nerve [53], FOXA2 in pancreas [54], etc. Interestingly, the similarity between tissues in terms of most relevant TFs in each tissue recapitulates, to a large extent, the developmental relationships between tissues. Our derived relationships among tissues separates endodermal organs derived from foregut (stomach and esophagus) from those derived from the hindgut (colon). While the separation of liver and pancreas from other endodermal lineages may seem discordant, this may be explained by potentially parallel development of these two tissues from the gut tube as recently suggested by single cell analysis [39]. Likewise, the two ectodermal lineages—skin and brain, are clearly separated from endodermal and mesodermal lineages. Apart from developmental links, our derived tissue relationship clusters the secondary female reproductive organs based on the structural proximity while separating out the primary reproductive organs (ovary and testis). Intriguingly, our model groups adipose tissue with breast, likely capturing the largely adipose content in the breast tissue.

It is tempting to speculate that the requirement for TRGs to be robustly expressed together in a tissue imposes some organizational constraints on their genomic location, because the genome is organized into TADs within which there is a greater spatial interaction [55] as well as greater co-expression [56]. These expectations bear out in our analysis showing that TRGs tend to occur in genomic clusters, in particular within TADs. While these tendencies are statistically significant, they are modest, suggesting that gene expression coordination among TRGs are far more complex and genomic organization is likely one of many factors affecting it.

Numerous previous studies, dating as far back as 1858 (Virchow's embryonal rest hypothesis), have observed parallels between cancer and development [57–59], proposing that cancer initiating cells are stuck at an earlier undifferentiated precursor developmental stage of the respective cell type, with activated immune suppressive and mesenchymal gene programs characteristic of early development [59]. This would suggest that genes with perturbed expression in cancer are likely to be involved in tissue development and function. Indeed, the most

tissue-relevant genes as prioritized by FUGUE is highly enriched for differentially expressed genes, either up or down, in the corresponding cancer.

Overall, we present the first integrative model–FUGUE, to prioritize tissue-relevant genes, apply FUGUE to prioritize genes in 30 human tissues, provide a software to apply our model to a novel context, show several interesting properties of TRGs—genomic clustering, association with cancer, and their ability to reveal tissue developmental and structural relationships.

## Methods

### Tissue-specific protein interaction networks

We used HIPPIE v2.2 protein interaction dataset [60] consisting of 16573 genes and 374108 interactions. To generate the tissue-specific networks, we retain an edge between genes g1 and g2 if both genes' expression is higher than their median (across all samples) gene expression in at least 25% of the samples in the particular tissue. In the tissue-specific networks, the number of genes (nodes) across all tissues ranged from 13526 to 15619 with a mean of 15115 and the number of interactions across all tissues ranged from 87259 to 342339 with a mean of 270823.

### Biological features

We consider the following biologically motivated transcriptomic and network-related tissue-specific features of a gene:

1. *Mean Z score.* A gene is expected to be expressed at a high level in relevant tissue relative to other tissues. We thus Z-transformed each gene's expression across samples, and for each gene-tissue pair, calculated the gene's mean Z-score across all samples of the tissue. Note that, even though we used the tissue-specific protein levels as a filter for our gold set (below), because of imperfect correlation between protein and RNA levels [61], as well as a limited number of samples in the protein atlas, the mean Z-score is not redundant, and provides independent information.

2. *Expression breadth.* We expect that the genes relevant to a tissue's function will be broadly expressed across individuals. Breadth was defined as the fraction of tissue samples in which a gene's expression was greater than its median expression estimated from all tissue samples.

3. *Mean Expression*. Mean expression across all the samples for a given gene in Tissue.

4. *Median Expression*. Median expression across all the samples for a given gene in a Tissue.

Multiple previous studies have linked expression variability or dispersion with gene function [62,63], motivating the choice of next three features.

5. *Median absolute deviation Median (MADM).* The median of absolute differential expression of the gene in a sample to the median expression of the gene across the samples.

6. *Standard Deviation*. Standard deviation of the expression of a gene across all tissue samples.

7. *Coefficient of Variation (CV)*. The ratio of standard deviation and mean expression.

The following features are calculated over tissue-specific protein interaction networks, and therefore tissue-specific.

8. *Centrality*. Earlier studies [64] have shown that essential genes tend to have high centrality in the protein interaction network. Centrality of a node is the fraction of all-pairs shortest paths that pass through the node.

9. *Clustering Coefficient*. This measures whether a gene is part of a tightly connected community of genes, and therefore likely to be functionally important. It is defined as the ratio of number of interactions between neighbors to the total number of such possible interactions between them.

10. *Degree (Number of neighbors).* Number of immediate neighbors of a gene. High degree nodes, termed hubs, are known to be essential for cellular functions.

11. *Number of Kinase neighbors.* Considering the importance of kinases as regulators, we measured for each gene the number of neighbors that are Kinases.

12. *Number of TF neighbors.* Analogous to Kinases, for transcription factors. These two features capture the complexity of regulation of a gene.

13. *Mean expression Z-score of neighbors.* This feature extends the z-score of the gene's expression to its neighbors. We reasoned that the neighbors of critical genes may also exhibit tissue-specific expression.

14. *Mean expression breadth of neighbors.* Analogous to the previous feature, but for expression breadth.

## Gold set curation

There are a total of 453455 Gene-Tissue (G-T) pairs (the gene set varies across tissues). A previous study has linked human diseases to specific tissues based on manual curation [9]. We integrated this tissue-disease mapping with disease-gene mapping in OMIM (omim.org) and HPO [10] compiled in a previous work [9] to identify 6812 Positive G-T pairs across 29 tissues. The number of Positive genes across the tissues ranges from 18 in Prostate to 1809 in Brain. The remaining 446643 G-T pairs are 'Unlabeled' by default, some of which may be positive. To identify a subset of the Unlabeled G-T pairs that are very likely to be non-functional, we relied on Human Protein Atlas (HPA) [11]. If the gene is not expressed in any of the HPA samples in the respective tissue, we consider the gene in that tissue to be a negative instance. This yielded 66250 Negative G-T pairs overall.

## Model building and prediction

Based on the aforementioned feature values computed for all G-T pairs, we implemented an XGBoost classifier which predicts the probability of a G-T pair being positive. XGBoost is robust even when features are highly multicollinear. Also, to make sure the model does not overfit and generalize well, we chose the trees to be relatively shallow. We used maximum depth of 3 with logistic loss as the objective. Since the ratio of Positives to Negatives is ~0.1, we weighted the objective to penalize Positive class misclassification 10 times compared to Negative class. We used 5-fold cross validation for 100 ensembles. To get an unbiased probability score for each G-T instance, we implemented a leave-one-out approach to build the model on the rest of the Positive and Negative G-T pairs and predict on the G-T instance in consideration.

## Genome wide association study (GWAS) validation

We obtained the associated genes for 551 traits having at least 5 genes from the GWAS catalog [23]. While there are multiple approaches used in the literature to estimate gene-centric

association scores, we followed a stringent approach where only the SNPs within the gene locus were used to associate a gene to a trait. A gene was linked to a trait if it harbored a SNP with association p-value < e-10. For all 30 tissues in our dataset, and for each trait, we assessed whether the trait-associated genes significantly overlap with the top 10% Z-score genes or top 10% of FUGUE genes or both with an odds ratio > 1.5 and p-value < 0.05. We only retained the GWAS genes for which a FUGUE score was available.

## Mouse knockout data validation

Mouse Genome Informatics database [65] provides anatomical phenotypes resulting from genome-wide gene knockout studies, thus linking genes to phenotypes. We curated a list of genes that when knocked out caused a disease/birth defect for 6 tissues—Heart, Breast, Muscle, Lung, Ovary and Testis. We got the human orthologs for these genes and compared the ranks of these genes relative to random expectation of 0.5.

## ESC KO validation

Previous genome-wide CRISPR-*Cas9* based gene knockout studies have quantified gene essentiality in haploid human pluripotent cells (hPSC) [22]. We obtained gene expression raw counts for 317 ESC samples from [21] and calculated the TPM values based on the transcript lengths. We then calculated the expression and network features of all genes. Using the 500 most essential genes [22] that are present in the ESC expression dataset [21], we checked if our independently trained model ranked the 500 genes higher than expectation.

## Identifying the developmental lineage of tissues from tissue-relevant genes

For the tissues from the endodermal, mesodermal and ectodermal lineage, we extracted the top 20 TFs [66] based on their FUGUE score. The distance between a pair of tissues was defined as "1—Jaccard similarity index" of the two sets of 20 TFs. The distance matrix was then subjected to hierarchical clustering (Euclidean distance using the Ward variance minimization algorithm) to infer a lineage tree for the tissues. A similar analysis was also done for the primary and the secondary reproductive organs to obtain the corresponding lineage tree.

## Genomic clustering of tissue-relevant genes

For a set G of genes, we define a cluster as a group of five (alternatively, three) or more consecutive genes in G separated by no more than 500 kb. We then calculated the fraction $f$ of genes in G that are included in such clusters. A higher $f$ indicates genomic clustering. We then generated a "null distribution" for $f$ based on 1000 samples of a random gene set (from a universe of 19201 genes obtained using the biomaRt package in R) of the same size as G, additionally controlling for the number of genes per chromosome. Based on the null distribution, the Z-score and the empirical p-value for $f$ were estimated. A Z-score > 0 indicates genomic clustering.

## Clustering of tissue-relevant genes within TADs

We obtained the genomic coordinates of TADs mapped to the hg19 genomic coordinates from [67]. For a gene set G, we computed the number of TADs that cover all genes in G—smaller this number, the greater the clustering within TADs. We compare this number against a "null distribution" generated as above, to estimate the Z-score and the p-value. A Z-score < 0 indicates clustering.

## Overlap between TRGs and DEGs

Based on a mapping between tissues in GTEx and cancer types in TCGA, for 19 cancer types, up- and down-regulated genes in each cancer type was obtained from [48]. For each tissue, we assessed whether TRGs were enriched (using Fisher test) for up- or down-regulated genes in the corresponding cancer. The odds ratio and p-value for each test was noted. An Odds-ratio > 1 indicates enrichment.

## Supporting information

**S1 Fig. For two features, the figure shows the mean value of the feature (y-axis) among the positive (P) and negative (N) genes in each tissue having positive and negative genes.** (TIF)

**S2 Fig. The figure shows the z-score ranks of the top 5% TRGs as ranked by FUGUE.** (TIF)

**S1 Data. Includes the positive, the negative, and the unlabeled genes in each tissue, along with their FUGUE scores.** (TSV)

**S2 Data. Enriched GO terms among the TRGs in each tissue, one tissue per sheet.** (XLSX)

**S1 Table. Gene counts and mean expression of negative, positive, and unlabeled genes in all tissues.** (XLSX)

**S2 Table. Mouse knockout genes with phenotypes in 6 tissues.** 191 genes whose germline deletion result in phenotypic aberration that could be unambiguously ascribed to one of the 6 tissues—Heart, Breast, Thyroid, Muscle, Lung, and Ovary. The tables show a few key feature values for these genes and the overall score for tissue relevance as computed by FUGUE. (XLSX)

**S3 Table. The table shows the GWAS traits revealed based on overlap between the top ranked genes by FUGUE or Z-score and the trait-associated gene.** For each tissue, the revealed traits are grouped into three groups colored Blue: Traits revealed by FUGUE alone, Orange: Traits revealed by Z-score alone, and Purple: Traits revealed by both. (XLSX)

**S4 Table. Top 20 TRTFs in each tissue ranked by FUGUE scores.** (XLSX)

**S5 Table. Clustering statistics.** GENES: number of genes in top 5% TRGs. F: Fraction of TRGs covered by clusters. F_r: Expected fraction. Std(F_r): Standard deviation of expected fraction. Z-SCORE(F): Z-score of F. P-value(F): Empirical p-value of F. (XLSX)

**S6 Table. TAD statistics.** #TAD: Number of TADs with at least 2 genes covering all top 5% TRGs. Other columns are self-explanatory. (XLSX)

**S7 Table. DEG statistics: Each table estimates the significance of overlap between TRGs and DEGs.** The top table considers all DEGs, the middle table considers only the DEGs upregulated in tumor relative to normal, and the bottom table considers only the genes

downregulated in tumor. Columns are self-explanatory.
(XLSX)

## Acknowledgments

The authors thank Drs. Shan Li, Vishaka Gopalan, Gulden Olgun, and Piyush Agrawal for providing comments on the initial draft.

## Author Contributions

**Conceptualization:** Sridhar Hannenhalli.

**Formal analysis:** Gowthami Somepalli, Sarthak Sahoo.

**Funding acquisition:** Sridhar Hannenhalli.

**Investigation:** Sridhar Hannenhalli.

**Methodology:** Gowthami Somepalli, Sridhar Hannenhalli.

**Software:** Gowthami Somepalli.

**Supervision:** Sridhar Hannenhalli.

**Validation:** Gowthami Somepalli, Sarthak Sahoo, Arashdeep Singh.

**Writing – original draft:** Gowthami Somepalli, Sarthak Sahoo, Sridhar Hannenhalli.

**Writing – review & editing:** Gowthami Somepalli, Sarthak Sahoo, Sridhar Hannenhalli.

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
