## [Decision Letter · Decision Letter 0]

3 Apr 2021

Dear Mr Sahoo,

Thank you very much for submitting your manuscript "Prioritizing and characterizing functionally relevant genes across human tissues" for consideration at PLOS Computational Biology.

As with all papers reviewed by the journal, your manuscript was reviewed by members of the editorial board and by several independent reviewers. In light of the reviews (below this email), we would like to invite the resubmission of a significantly-revised version that takes into account the reviewers' comments.

We cannot make any decision about publication until we have seen the revised manuscript and your response to the reviewers' comments. Your revised manuscript is also likely to be sent to reviewers for further evaluation.

Sincerely,

Zhaolei Zhang

Associate Editor

PLOS Computational Biology

Jian Ma

Deputy Editor

PLOS Computational Biology

Reviewer's Responses to Questions

**Comments to the Authors:**

Reviewer #1: In this paper, the authors intend to predict the genes that are likely relevant to a tissue’s function, which is defined as inactivation of the gene has a phenotypic effect observable at the tissue level. The authors tested 14 features and then used them in a machine learning model called FUGUE to predict tissue relevant genes. The authors validated the predictions by examining the genes’ functions and location on the genome.

The work seems to be significant because most current research focus on either cellular level or organism level, while research on the relevance on the tissue level is lacking. However, there are major concerns that need to be addressed before the paper can be accepted.

1. The authors’ proposal to use genes that don’t express in certain tissues as the negative training set seems problematic. Can the authors provide information on how many genes are expressed and not expressed in each type of tissues? The most meaningful predictions should be to differentiate relevant genes from all genes that are expressed in a tissue. Using genes that don’t express in a tissue might cause the algorithm to recognize features that are not truly responsible for the “relevance” of the genes.

2. It is a little surprising to see among all 14 features, 12 achieves statistical significance to differentiate positive and negative sets. Can authors provide biological meanings on why these features work? Most of the features are related to gene expressions, but in negative training sets, the gene expression is essentially 0 since they don’t express in the tissues. Could this be the reason why features based on gene expression all work well?

3. I don’t see the authors list the number of TRGs predicted for each issue. Perhaps the authors could provide this information?

Reviewer #2: Somepalli et al developed a machine learning model – FUGUEO, aiming to predict tissue-relevant genes. They combined 14 features in a XGBoost (gradient boosted

decision tree) model using data either from tissue-specific gene expression or tissue-specific

protein interaction networks. They demonstrated that FUGUE can discriminate

between evidence-based known positive and negative gene-tissue pairs, with an average auROC score of 0.86 and auPRC of 0.50, based on 5-fold cross validation. Moreover,

they evaluated the performance by analyzed independent datasets from 1) a list of known genes relent to phenotypic aberration in relevant tissues and 2) gene expression data generated from 317 human Embryonic Stem Cell (hESC) transcriptome with additional informaiton of the 500 essential genes from knock-out experiments in hESC. Overall, they provided additional evidence to support the tissue-relevant genes prioritized based on FUGUE. For the downstream analyses, they additionally showed that the identified tissue-relevant genes may be relent to tissue development, structure, and cancer oncogenesis.

Overall, I think this is a well-conducted study with reasonable motivation to address the challenge of identifying tissue-relevant genes. The developed FUGUE tool could be useful in the related filed. However, I have specific comments/concerns below:

1) When they applied FUGUE to the data from hPSC, They found that a conventional approach Z-score ranks of the essential genes was higher than the FUGUE ranks. This could be a major limitation in their approach. It’s still unclear whether FUGUE may outperform previous approaches in various conditions, while further discussion are needed.

2) The authors analyzed the data from Topologically Associated Domains (TADs) and showed additional evidence that the identified tissue-relevant genes were enriched in TADs. However, the findings did not really provide a direct evidence to support their hypothesis. They should analyze tissue-specific enhancers which have been released the Roadmap project or the most recent literature from the EpiMap project.

3) They mapped to genes from GWAS-identified loci. It’s unclear how to link target genes for GWAS-identified variants (i.e. eQTL analysis in specific tissues or just location-based analysis). Should clarify this.

4) In Figure 4, it will be helpful to provide the information tissue-relevant transcription factors to facilitate the review.

Reviewer #3: In this manuscript the authors applied a machine learning method FUGUE to identify the genes that are highly tissue-specific. The model applies features from transcriptomic data and protein interaction network. They achieve quite a remarkable cross-validated auROC score and the results are replicated in independent datasets. They also provide ranked lists of these tissue relevant genes (TRGs) in 30 human tissues as well as they offer an open-source software application.

In my opinion this is a well written manuscript, clearly depicting the goals of the study and walking the reader throughout the entire journey from the limitations of the previous studies, to their novel approach and the data limitations that impact the interpretations of their results. The methods applied are rigorous and fully support the conclusions.

Fig 1. Perhaps some revisiting of this figure to make it more readable. It took me a bit to read the caption and follow the arrows to understand the process. I’d suggest perhaps to give more of a top-bottom or left-right flow.

Fig 2A: what tissue has the average positive score very close to the negative one?

KRAS is an important gene in many cancers, but often in colorectal adenocarcinoma. Was there any association in addition to ovarian?

Data_File_2: “Table S1” is labeled as Table S2 in the gray box

**Have all data underlying the figures and results presented in the manuscript been provided?**

Reviewer #1: **No: **

Reviewer #2: Yes

Reviewer #3: Yes

PLOS authors have the option to publish the peer review history of their article (what does this mean?). If published, this will include your full peer review and any attached files.

Reviewer #1: No

Reviewer #2: No

Reviewer #3: No
---

## [Decision Letter · Decision Letter 1]

17 Jun 2021

Dear Mr Sahoo,

We are pleased to inform you that your manuscript 'Prioritizing and characterizing functionally relevant genes across human tissues' has been provisionally accepted for publication in PLOS Computational Biology.

Best regards,

Zhaolei Zhang

Associate Editor

PLOS Computational Biology

Jian Ma

Deputy Editor

PLOS Computational Biology

Reviewer's Responses to Questions

**Comments to the Authors:**

Reviewer #1: The authors have addressed my comments.

Reviewer #2: The authors have adequately addressed my comments.

Reviewer #3: The authors successfully addressed my questions.

**Have the authors made all data and (if applicable) computational code underlying the findings in their manuscript fully available?**

Reviewer #1: Yes

Reviewer #2: Yes

Reviewer #3: Yes

PLOS authors have the option to publish the peer review history of their article (what does this mean?). If published, this will include your full peer review and any attached files.

Reviewer #1: No

Reviewer #2: No

Reviewer #3: **Yes: **Andrea Sboner

---

## [Editor Report · Acceptance letter]

2 Jul 2021

PCOMPBIOL-D-21-00265R1 

Prioritizing and characterizing functionally relevant genes across human tissues

Dear Dr Sahoo,

I am pleased to inform you that your manuscript has been formally accepted for publication in PLOS Computational Biology. Your manuscript is now with our production department and you will be notified of the publication date in due course.

With kind regards,

Olena Szabo
